# Design of Evaluation Classification Algorithm for Identifying Conveyor Belt Mistracking in a Continuous Transport System’s Digital Twin

**DOI:** 10.3390/s24123810

**Published:** 2024-06-13

**Authors:** Gabriel Fedorko, Vieroslav Molnar, Beata Stehlikova, Peter Michalik, Jan Saliga

**Affiliations:** 1Faculty BERG, Technical University of Kosice, Park Komenského 14, 040 01 Kosice, Slovakia; beata.stehlikova@tuke.sk; 2Faculty of Manufacturing Technologies, Technical University of Kosice with a Seat in Presov, Bayerova 1, 080 01 Presov, Slovakia; vieroslav.molnar@tuke.sk (V.M.);; 3Faculty of Electrical Engineering and Informatics, Technical University of Kosice, Letna 9, 040 01 Kosice, Slovakia; jan.saliga@tuke.sk

**Keywords:** belt mistracking, digital twin, transport system, algorithm

## Abstract

A prerequisite for continuous transport systems’ operation is their digital transformation, which interprets operating conditions based on the availability of a wide range of data and information in the form of measured quantities that can be obtained, for example, by experimental measurement. To implement digital transformation in continuous transport systems, it is necessary to examine and analyze the informative value of individual measured quantities in detail. Research in this area must focus on identifying addressable quantities with a clear, informative value. Such an approach enables the monitoring of continuous transport systems operation and performance of operational diagnostics, the objective of which should be identifying undesirable operating conditions. Within this paper, research will be presented aiming to verify the hypothesis that, based on a measurement of selected parameters, it is possible to identify belt mistracking in a continuous transport system. Belt mistracking is an undesirable condition that can cause a conveyor belt to converge and thus seriously turn off an entire transport system. The research results confirmed the established hypothesis. Based on this, an evaluation algorithm was created for on-time evaluation. The proposed algorithm is also suitable for the needs of a digital twin of a continuous transport system.

## 1. Introduction

Continuous transport systems have a transport capacity that enables the transport of large volumes of different material categories. This is one of the main criteria predetermining them for deployment within various technological processes. Individual types of continuous transport devices can be classified based on several aspects. One is whether they use a traction element to secure transport or whether it is secured without it. Conveyor systems with a traction element consist of large devices, among which belt conveyors dominate.

In general, belt conveyor operation is not tricky, but it requires, among other things, high-quality implementation of regular maintenance and monitoring [1]. Thanks to this, belt conveyors feature operational and technological reliability [2], which is also significantly reflected in their efficiency and operating costs. One of the areas on which the monitoring of continuous transport systems should focus is the identification of undesirable operating conditions [3].

An undesirable operating condition of a continuous transport system is when the transport process is jeopardized, can be interrupted, or its structural parts may be damaged. Online monitoring can prevent such undesirable conditions using measured selected parameters and operating indicators [4].

This issue is not new; several authors have addressed it in the past [5,6]. Various methods can be used for online monitoring, such as non-contact measurement [7], the sound-based method [8], laser scanning technology [9], or sound and thermal infrared image features [10]. All measurement methods applied in this way can be used for the online monitoring of continuous transport systems [11] and determining their structural parts’ condition, mainly the conveyor belt [12].

The application of measurement for the needs of continuous transport systems should, among other things, lead to the creation of intelligent transport systems [13] to ensure fast and accurate diagnosis [14] using appropriate methods and approaches for their reliable and safe operation [15]. The mentioned approach must be gradually improved to include different undesirable operating conditions [16]. Currently, regarding belt conveyors, knowledge is available about damage to a conveyor belt [17,18], transmission [19], or its other structural parts, such as rollers in the idler housing [20]. This knowledge remains insufficient, however, and further research must be conducted to analyze and understand other undesirable operating conditions, such as belt mistracking [21].

Belt mistracking is an issue that has been addressed in detail by several researchers [22,23,24], but it has also been paid attention to by manufacturers of conveyor belts and devices [21,25]. Otto and Katterfeld [26] reported that belt mistracking is one of the critical operational problems of a belt conveyor. Practically, this is the most dangerous and costly failure of belt conveyors. The reasons for its occurrence and countermeasures are known; however, the cause of mistracking and its effect on the belt can only be predicted based on empirical experience. Until now, a satisfactory simulation method has not existed due to the enormous size of the belt conveyor and its nonlinear deformation with many contact problems. Dabek et al. [27] compared the mechanical and operational parameters of the conveyor’s electric drive on a laboratory test rig with a problem of lateral mistracking. The mistracking triggered a sudden, differentiated increase in resistance against the machine’s movement, directly affecting the recorded values of forces in the drive unit and tensioning mechanism. It increased the electric power consumption of the drive motor. Monitoring belt movement based on these parameters is inaccurate because many other factors along the conveyor path are potentially responsible for increased recorded currents and forces.

Based on the mentioned facts, research was carried out, the results of which we present in this article. Within this research, based on experimental measurements, a decision-making algorithm was searched for that would be able to identify belt mistracking emergence. The algorithm was researched regarding its application in the digital twin method. It is a proven and robust method used in various systems for efficient operation and optimization [28]. The digital twin method can also be applied to the needs of belt conveyors [29].

The presented results contribute to the digital transformation of continuous transport systems. The results support research on effective management based on data collection and evaluation. This paper can be considered a guide on how digital transformation with the support of online measurement can be implemented. It presents approaches that have general validity. They can be used to monitor and diagnose a whole set of other operating conditions. This will significantly increase their operational management efficiency and reliability. This paper introduces the issue of implementing the digital twin method in continuous transport systems. This paper also presents the general concept of a digital twin of a continuous transport system. The mentioned concept enables the use of decision-making algorithms for the effective management of the operation of continuous transport systems.

## 2. Materials and Methods

Belt mistracking research is not easy. We can approach its implementation by analyzing the causes of its occurrence and the consequences manifested on the conveyor belt. The second aspect that we can use is research aimed at the early identification of belt mistracking during the operation of continuous transport devices. In both approaches, experimental measurements and a detailed evaluation are necessary.

Various specially designed test rigs can effectively be used for experimental measurements (Figure 1 and Figure 2).

With the mentioned approach, it is thus possible to obtain valuable data that analyze the process and consequences of belt mistracking in detail. However, research on belt mistracking should lead to a search for a method that allows its identification while the continuous transport device is in operation. A digital twin is an efficient method for measuring results.

To research the possibility of identifying and quantifying the asymmetry of the belt’s tensioning, repeated experimental measurements were carried out on a test rig used to test the properties of rubber textile conveyor belts. The principal scheme of the test rig is in Figure 3.

The test rig consists of a stand with three hexagonal idler housings, No. 1, No. 2, No. 3, and a transport belt. In the lower part of the test rig, the conveyor belt unfolds and is fixed using a tension plate, which tensions it to a required value with two tension screws in ID23 and ID24. These tension screws, in addition to contributing to uniform, symmetrical conveyor belt tensioning, enable various states of asymmetric tensioning. Strain gauges continuously measure the size of tension forces (TFs) in ID23 and ID24. In the next part of the test rig, the conveyor belt is gradually folded into an overlapped U shape. The size of reactions to the conveyor belt’s folding into a U shape is continuously measured using two rollers with measuring strain gauges ID19 and ID20. In the central part of the test rig, a gradual folding of the conveyor belt into a circular shape with an overlap follows. The size of reactions to keep the conveyor folded and overlapped is continuously measured using 18 rollers with measuring strain gauges ID1–ID18, located in the three idler housings, No. 1, No. 2, and No. 3. In the upper part of the test rig, the conveyor belt is folded and of a circular shape.

For computation, graph plotting, and statistical evaluation, Microsoft Excel and the R-statistics environment [30] were utilized. The employed methods were the following: radar chart, box plot, scatter plot, line chart, descriptive statistics, analysis of variance, Hartley’s Fmax test, Shapiro–Wilk test of normality, Tukey’s multiple comparisons of means, The F-test for evaluating the statistical significance of the model, and R-squared criterion, also known as the coefficient of determination.

Analysis of variance (ANOVA) is a statistical method that was developed by R.A. Fisher to test differences between two or more groups. The purpose of ANOVA is to determine if there are any statistically significant differences between the means of independent groups [31]. The assumptions for ANOVA are the homoscedasticity and normality of residuals. The term “homoscedasticity” means that the variances within each group should be approximately as equal, while “normality” means that the residuals are the deviations from the model data; in the context of ANOVA, the model data are considered the means of each group. The Hartley test was used to confirm the assumption of homoscedasticity. To confirm normality, the Shapiro–Wilk test was used. Tukey’s honestly significant difference (HSD) test is a post hoc analysis used after performing an ANOVA. It controls for Type I error across multiple comparisons and identifies pairs of group means that are significantly different.

The F-test is commonly used to evaluate the overall significance of a regression model, and the R-squared criterion measures the proportion of the variance in the dependent variable that is predictable from the independent variables [32].

### Digital Twin

With the development of digital transformation, the “Digital Twin” method is increasingly being used in various fields. It is a highly efficient method based, among other things, on online measurements of multiple values and their follow-up assessment. Based on the above, the process is also suitable for application in continuous transport systems [33].

The principal structure of the digital twin (Figure 4) is essential when using it, and it must be designed according to the relevant device. A digital twin can be applied to any continuous transport device.

Continuous transport devices’ primary digital twin concept comprises hardware and software components [34]. The hardware component is represented by sensors for measuring individual monitored parameters, technical support for its operation (computer technology), and actuators for implementing the transformation of commands from the digital twin to its real physical object, which the digital twin represents. The software component is the software in which the physical object model is created. In addition to creating a digital version of the object, this software enables processing, evaluation, or optimization based on measured input data.

The measured data evaluation is one of the critical functionalities of the digital twin, as the result being sent to a physical object is formed on its basis.

## 3. Undesirable Operating Conditions of Continuous Transport Systems

Undesirable operating conditions of continuous transport systems can be classified by their nature as the following:(a)dynamic;(b)static;(c)combined.

Undesirable operating conditions from the dynamic category are mainly manifested during the operation (running) of a continuous transport device. They do not manifest when they stop and are more challenging to identify. In this category, for example, the convergence of a conveyor belt is included, which can be caused by an asymmetric position of the drive drum or damage to the conveyor belt’s internal structure. One or a combination of several factors can trigger these cases.

The category of undesirable operating conditions of a static nature primarily includes mechanical damage to the structural parts of a continuous transport system. Specifically, this includes defects and damages to the conveyor belt, the frame of the transport device, or the idler housings. The mentioned defects also persist when the transport system stops.

The last category includes undesirable conditions, regardless of whether the continuous transport device is in operation or stopped. For example, the rollers in the idler housing are missing and the conveyor belt is being damaged during operation, while during shutdown, the belt is in an undesirable position.

### 3.1. Identification of an Undesirable Operating Condition

Identifying an undesirable operational state of a continuous transport system during its operation is difficult. Currently, several identification mechanisms are based on various non-destructive approaches. Their most common feature is that the control of continuous transport system operation takes place at one or more control points, and the transport device condition is evaluated afterward. The disadvantage of this approach is that outside these points, the transport system’s monitoring is sporadic or insufficient, and the resulting malfunction is identified only after some time. This can ultimately result in severe damage to the transport system. Belt mistracking is among the undesirable operating conditions of continuous transport systems, which, due to late identification, can seriously damage an entire transport system.

#### Causes of Belt Mistracking

Belt mistracking is a well-known phenomenon and an undesirable operating condition of continuous transport systems. It is widespread in classic belt conveyors. The causes of its formation are in Figure 5.

Its identification during the operation of a continuous conveyor system is more complex. Still, simple mechanical devices exist to prevent damage, which put the conveyor belt back to its original position.

This undesirable operational phenomenon also occurs in other continuous transport systems, such as pipe conveyors. Belt mistracking is a hazardous condition for the operation of pipe conveyors, which can have fatal consequences due to their construction. It can damage the conveyor belt entirely and disrupt the entire conveyor’s operation.

## 4. Identification of Belt Mistracking by an Evaluation Algorithm

The identification of belt mistracking is complex, and the consequences of it can be severe. Therefore, this undesirable condition needs to be identified at an early stage to avoid potential damage and operational interruption of a continuous transport device. Based on available knowledge, a scientific hypothesis was concluded that belt mistracking could be identified in time based on identified asymmetry in the tensioning of a continuous device’s conveyor belt.

Our experiment aimed to determine a method for identifying and quantifying the asymmetry of conveyor belt tensioning. This primary objective comprised two concurrent sub-objectives:To select the optimal position(s) from the various positions where contact force was measured, such that the asymmetry of tensioning could be estimated using a regression model;To establish the best method for handling the variable “asymmetry of conveyor belt tensioning”, including determining how to construct this variable (whether as a ratio or a difference) and, consequently, how to quantify the asymmetry.

### 4.1. Experimental Settings

The experimental measurements were carried out on a rubber textile conveyor belt EP 500/3 4 + 3D. The operating conditions determined a tension force of approximately 40,000 N for the selected rubber textile conveyor belt. Table 1 shows the settings chosen for researching the tensioning asymmetry of the conveyor belt without transported material in the test rig.

Calculations were carried out using the arithmetic mean of 15 consecutive TF measurements after the tensioning finished; the measurement frequency was 0.1 s.

### 4.2. Processing of Measured Data

Figure 6 shows the processing of measured data from repeated experiments when estimating the asymmetry of the transport belt tensioning in the test rig without material.

In Step 1, the force ratios at the measured positions of the rollers with measuring strain gauges ID1–ID20 in the test rig were displayed graphically, resulting in the first selection of suitable positions for the rollers with measuring strain gauges.

Step 2 encompassed determining the variables to be used to define the asymmetric tensioning.

In Step 3, the parameters of the designed regression model were estimated, which would then be verified with statistical indicators and independent test data. Then, the limits of safe asymmetry were determined, and an algorithm for classifying the asymmetry was designed. This classification algorithm was then verified with independent test data.

#### 4.2.1. Step 1—Force Ratios of CF during Asymmetric Tensioning

In Step 1, graphical CF curves were analyzed at the rollers’ positions with the measuring strain gauges ID1 to ID20. Figure 7, Figure 8, Figure 9 and Figure 10 show the measured values of the CF for the set value of TF = 40,000 N. The roller’s position with the measuring strain gauge for asymmetric classification suitability was determined by a sufficient (visible) difference in the measured CF values for different levels of the set asymmetry of the TF. The CF range was set as the maximum CF value for each idler housing in Figure 7, Figure 8 and Figure 9. Figure 7 shows that the CF range for idler housing No. 1 was 400 N.

Figure 7 shows that in idler housing No. 1, the variability in CF at the positions of the rollers with measuring strain gauges ID1 to ID6 was insufficient to classify asymmetry for all levels of the set TF asymmetry.

Figure 8 shows the range for idler housing No. 2 at 120 N. It also shows that in idler housing No. 2, the variability in CF at the positions of the rollers with measuring strain gauges ID7 to ID12 was insufficient to classify asymmetry for all levels of the set TF asymmetry.

Figure 9 shows the range for idler housing No. 3 at 120 N. It shows that in idler housing No. 3, the variability in CF at the positions of the rollers with measuring strain gauges ID13 to ID18 was insufficient to classify asymmetry for all levels of the set TF asymmetry.

Figure 10 shows the AsymRatio quantity, which had values from 0.89 to 1.12, and the CF values on ID19 and ID20, which had a maximum value of 22.6 N. The AsymRatio, i.e., the share of ID23/ID24, was multiplied 10× to achieve approximately the same values in the graph scale. Figure 10 clearly shows that at the point of the conveyor belt folding into a U shape, the CF variability at the rollers’ positions with measuring strain gauges ID19 and ID20 was sufficient to classify asymmetry for all levels of the set TF asymmetry. Figure 7, Figure 8, Figure 9 and Figure 10 imply that only at the positions of the rollers with measuring strain gauges ID19 and ID20 could the difference in CF values be identified for individual levels of the set TF asymmetry.

#### 4.2.2. Step 2—Method of Defining Asymmetric Tensioning

This research step was carried out to find a way to define and estimate asymmetry, while its definition was based on CF variability. To investigate the CF variability and decide on a form of indicator to classify asymmetry, Table 2 was created, listing the values of CF [N] for all levels of the set asymmetry at TF = 40,000 N. Two tensioning screws at ID23 and ID24 tensioned the conveyor belt to the TF value (Figure 3). The CF was given for roller positions with measurement strain gauges ID1 to ID20. To provide an overview of the measured CF results’ variability for individual levels of set asymmetry, the minimum (min), maximum (max), range, average, and ratio statistical indicators were determined. The values in Table 2 confirmed an assumption that to classify asymmetry, the best positions were the rollers with measuring strain gauges ID19 and ID20. The values ratio = range/average was 85% for ID19 and 95% for ID20.
(a)Verification of the Applicability of the TF23/TF24 Share in ID23 and ID24 for Asymmetry Classification.

The tension forces in ID23 and ID24 fluctuate around a set value under actual test rig conditions, so describing the level of asymmetry by TF difference or a categorical variable may not always be the best choice. At the same time, under natural conditions, we do not assume the arising asymmetry is identical to the set difference between tensioning forces (TFs) in ID23 and ID24. That is why the known TF23/TF24 ratio at ID23 and ID24 was used to describe asymmetry and its levels.

**Research Hypothesis:** 
*The variable created as the ratio of TF23/TF24 is applicable for classifying the asymmetry of tensioning.*


The level of set asymmetry was chosen as a categorical variable (Table 1), which limited the applicability of its classification tools. Therefore, it was transformed into the numerical variable AsymRatio, determined by the ratio of TF23/TF24.

#### 4.2.3. Statistical Verification of the Hypothesis

Our research established the following hypothesis: Mean AsymRatio values are statistically significantly different for each pair of asymmetry categories. The verification of this hypothesis was conducted in two steps.

In the first step, using the ANOVA test (one-factor analysis of variance) [31], we confirmed that a pair of asymmetry categories existed for which the difference in the mean values of the AsymRatio variable was statistically significant, with a *p*-value of 2.2 × 10^–16^.

In the second step, using the Tukey HSD criterion [35], we confirmed a statistically significant difference in the average values of the AsymRatio for each pair of asymmetry categories in the pairwise comparison of all pairs. As shown in Table 3, the *p adj* values were less than 0.05, confirming the research hypothesis.

The verification of assumptions for the use of the ANOVA test was conducted as follows.

Hartley’s Fmax test was conducted to examine the homogeneity of variances among the groups [36]. The calculated test statistic was 14.261. The critical value for five groups and a minimum of five measurements per group at the 0.05 significance level was 16.3. Since our test statistic was less than the critical value, we failed to reject the null hypothesis of equal variances. Therefore, we do not have sufficient evidence to conclude that the variances among the groups were significantly different. Verification of the normality assumption for the residuals was performed using the Shapiro–Wilk test [37], resulting in a *p*-value of 0.5867.

The assumptions were met.

Figure 11 implies that the values of the AsymRatio variable differed statistically significantly for all levels of set asymmetry. The AsymRatioZero variable was derived from the AsymRatio variable by subtracting one from the TF23/TF24 ratio. The variable’s advantage is that it displays the asymmetry direction according to the signs + or − without further processing.

The ANOVA test and Tukey HSD results were also applied to the AsymRatioZero variable, created by subtracting one from the AsymRatio variable.
(b)Finding a Combination of CF Variables in ID19 and ID20 to Estimate the Ratio of TF23/TF24 in ID23 and ID24

A pair of variables, *U_ratio_* and *U_norm_*, was chosen to predict the AsymRatioZero:(1)$$U_{ratio}=\frac{\mathrm{CF}19}{\mathrm{CF}20}\;$$
where *U_ratio_* is the ratio of contact forces CF19 and CF20 in ID19 and ID20, while
(2)$$U_{norm}=\frac{\mathrm{CF}19-\frac{\left(\mathrm{CF}19+\mathrm{CF}20\right)}{2}}{\frac{\left(\mathrm{CF}19+\mathrm{CF}20\right)}{2}}$$

The expression in the numerator of variable *U_norm_* ensures that the asymmetry direction can be identified according to the sign in the *normID19* variable.

#### 4.2.4. Step 3—Classification and Verification

Two-step Classification and Verification:In the first step of classification and verification, the parameters of the regression model (3) were estimated using the method of least squares, which calculated the theoretical values of the variable AsymRatioZerofit (ARZf) for the known values of CF19 and CF20, corresponding to the variable AsymRatioZero. Then, the regression model (3) was verified using statistical indicators, comparing theoretical and empirical values and independent test data.

Estimation of the regression model parameters (3) using the method of least squares *a*_0_, *a*_1_, *a*_2_:(3)$$AsymRatioZerofit=a_{0}+a_{1}\cdot U_{norm}+a_{2}\cdot U_{ratio}$$

*a*_0_ = −0.068; *a*_1_ = 0.089; *a*_2_ = 0.028

Table 4 shows the regression model verification, using statistical indicators, and the evaluation of the following model parameters’ statistical significance: Estimate, Std. Error, t-value, and Pr(>|t|).

All estimated parameters *a*_0_, *a*_1_, *a*_2_ were statistically significant. The adjusted R-squared criterion value was 0.947, meaning the AsymRatioZero variable’s model data described 94.7% of the original AsymRatioZero data. The F-statistic value represents the F-criterion for evaluating the statistical significance of the model, 251.2 on 2 and 26 DF, and the *p*-value < 2.2 × 10^−16^, which means that the regression model was suitable.

Figure 12 shows the regression model verification by comparing theoretical and empirical values, which makes it clear that the estimate of the AsymRatioZero variable’s values was excellent. The horizontal dashed lines plot the set asymmetry levels for 0, 1, –1, 2, and –2, according to Table 1 (Asym).

The independent test data to verify the regression model were obtained as follows. For individual levels of set asymmetry and set TF, a moment in time was chosen randomly, and the TF and CF were read from it, resulting in a file with 94 CF and TF values. First, the theoretical AsymRatioZerofit values were calculated and compared with the actual values. The accuracy of the regression model of the independent data was given by the R-squared value of 0.984. The mean square error referred to as MSE Squared was 0.000165243, and the MSE calculated as the square root of the MSE Squared was 0.0129. The reliability of the AsymRatioZerofit variable estimate was sufficient. Figure 13 shows the original and computed values of the variable AsymRatioZerofit and the set asymmetry Asym levels.

Figure 13 shows that for the set asymmetry level asym_0, the values of AsymRatioZero and AsymRatioZerofit are very closely clustered around the zero value. For the set asymmetry levels asym_1 and asym_-1, the closeness of the AsymRatioZerofit values to AsymRatioZero was sufficient. Still, at the set asymmetry levels asym_2 and asym_-2, the model values were already far from the empirical values. However, this condition is not a problem, as in real situations, the purpose is not asymmetric tensioning but, contrary to correction of the asymmetric tensioning condition by adjusting the tension forces if classified asymmetry is larger than the permitted limit.
2.In the second step of classification and verification, the limits of safe asymmetry were first defined, then a classification algorithm was designed. This algorithm evaluates a necessary correction of the tension forces’ asymmetry for continuously entered values of the variable AsymRatioZerofit (ARZf). Then, the draft classification algorithm is verified using independent test data in a classification table.

Since the safe limit of set asymmetry was not determined in advance, the limit of set asymmetry ±1000 N was chosen. It was the set difference between TF23 and TF24 in positions ID23 and ID24. Based on the calculated value of AsymRatioZero, a safe asymmetry interval was estimated. Then, the classification in Table 5 evaluated whether the empirical values of AsymRatioZero, for which the crossing of a safe asymmetry limit was classified, corresponded to the classifications based on the ARZf variable’s theoretical values on exceeding the safe limit of asymmetry.

The limits of the safe asymmetry interval were determined according to (4), (5):(4)$$\bar{AsymRatioZero_{asym\_-1}}+3\cdot \sigma_{AsymRatiZero\_asym\_-1}$$
(5)$$\bar{AsymRatioZero_{asym\_1}}-3\cdot \sigma_{AsymRatiZero\_asym\_1}$$
where

$\bar{AsymRatioZero_{asym\_-1}}$ is the average of the variable AsymRatioZero for the asymmetry asym_-1

$\bar{AsymRatioZero_{asym\_1}}$ is the average of the variable AsymRatioZero for the asymmetry asym_1

$\sigma_{AsymRatiZero\_asym\_-1}\;$ is the standard deviation of the AsymRatioZero variable for the asymmetry asym_-1

$\sigma_{AsymRatiZero\_asym\_1}\;$ is the standard deviation of the AsymRatioZero variable for the asym_1 asymmetry

The values of the variable AsymRatioZero in the interval –0.0369 and 0.0344 determined the limits of safe asymmetry.

Subsequently, an evaluation classification algorithm for belt mistracking measurement was designed using the variable ARZf in Figure 14.

This was followed by verifying the evaluation classification algorithm in Figure 14, using independent test data (different from the data used to create the model in Step 1) in a classification table.

It follows from Table 5 that 91 out of 94 asymmetry cases were classified correctly, i.e., 96.8%. In three cases, the limit of safe asymmetry was misclassified. By checking the measured data in their vicinity, we found that after 0.5 s, the CF values in ID19 and ID20 changed. With online classification, the information about exceeding the interval of safe asymmetry would be available with less than a second delay, which is considered acceptable.

The evaluation classification algorithm was created and verified. The verification results proved the suitability of its practical application. Setting the exact values of a safe asymmetry limit must be resolved individually for a specific continuous transport system and corresponding transported material filling.

## 5. Conclusions

The digital twin method in continuous transport systems significantly increases operational reliability and safety. The digital twin has a perspective for use in various areas associated with continuous transport systems. One of the main conditions for its effective functioning is the existence of decision-making procedures that enable constant transport systems within various logistic processes. With a digital twin, it is possible, among other things, to identify undesirable operating conditions.

This paper presents research that resulted in the design of an evaluation algorithm for experimental belt mistracking measurement for the digital twin of a continuous transport system. As a damage indicator, the measurement of asymmetry during the tensioning of a conveyor belt was verified. For its diagnosis, a place for CF measurement and a method of determining asymmetry using the AsymRatioZero variable was proposed.

The research result of this paper is a verified algorithm for the digital twin method. Based on indirect measurement, it can continuously evaluate whether or not a correction of the symmetry of tensioning forces is necessary. To develop it, suitable positions of CF measurements, variables, regression model parameters, and limits of safe asymmetry were required.

The presented results (calculated classifier values, classification algorithm) were fixed to and their address derived for the device on which they were measured. The results guide the classification of asymmetric tensioning and correct it well if a digital twin is used for a specific continuous transport system. The limitation of this method is that each continuous transport system represents a unique device in terms of operating conditions and criteria characterized by particular properties.

Based on the knowledge obtained, further research will examine several levels of filling of a conveyor belt with material, and an evaluation algorithm will be added to the functionality of the digital twin, which will include the transported material.

In the future, the authors plan to repeat experiments for several levels of conveyor belt filling under different operating conditions of the belt conveyor. This research will be carried out not only on static but also on dynamic test equipment. The benefit for both science and practice will be the gradual linking of knowledge from previous measurements from individual experiments, realized on model devices (static and dynamic), such as correlations between measured characteristics and operating conditions, changes in the size of contact forces, and identification of other quantifying quantities. All planned steps in the scientific research will be carried out with the intention that the digital twin provides a reference to the condition/status of the conveyor belt running/operation, using well-accessible measurement positions, which, based on direct or indirect measurements, will provide information on the effect resulting from the changed operating conditions, such as an exceeded level of conveyor belt tension asymmetry, missing material, a missing roller in the roller chair, and other types of undesirable operating conditions.

## Figures and Tables

**Figure 1 sensors-24-03810-f001:**
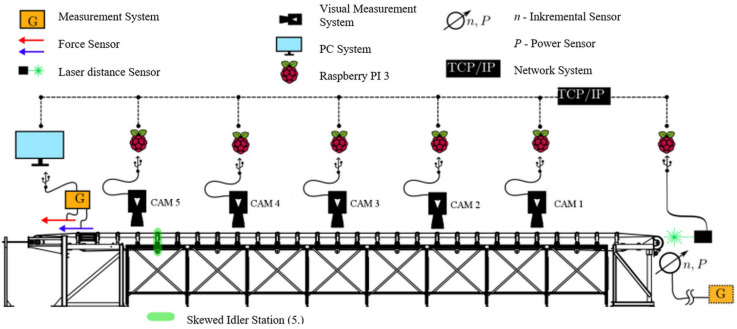
Sketch of the test rig at the University of Magdeburg (Magdeburg, Germany) for belt mistracking [24].

**Figure 2 sensors-24-03810-f002:**
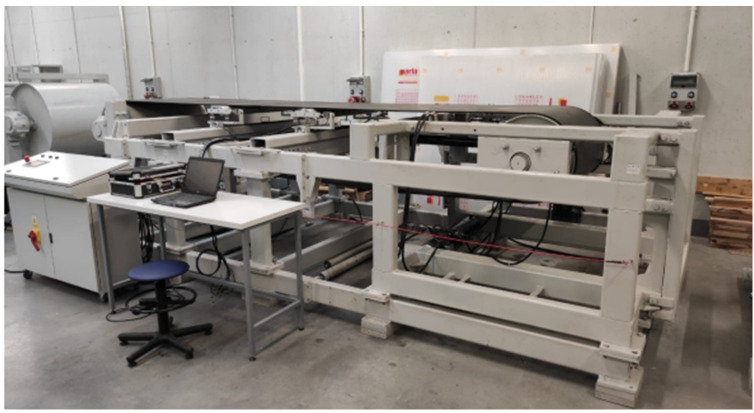
Laboratory test rig at the Wrocław University of Science and Technology (Wroclaw, Poland) for belt mistracking [27].

**Figure 3 sensors-24-03810-f003:**
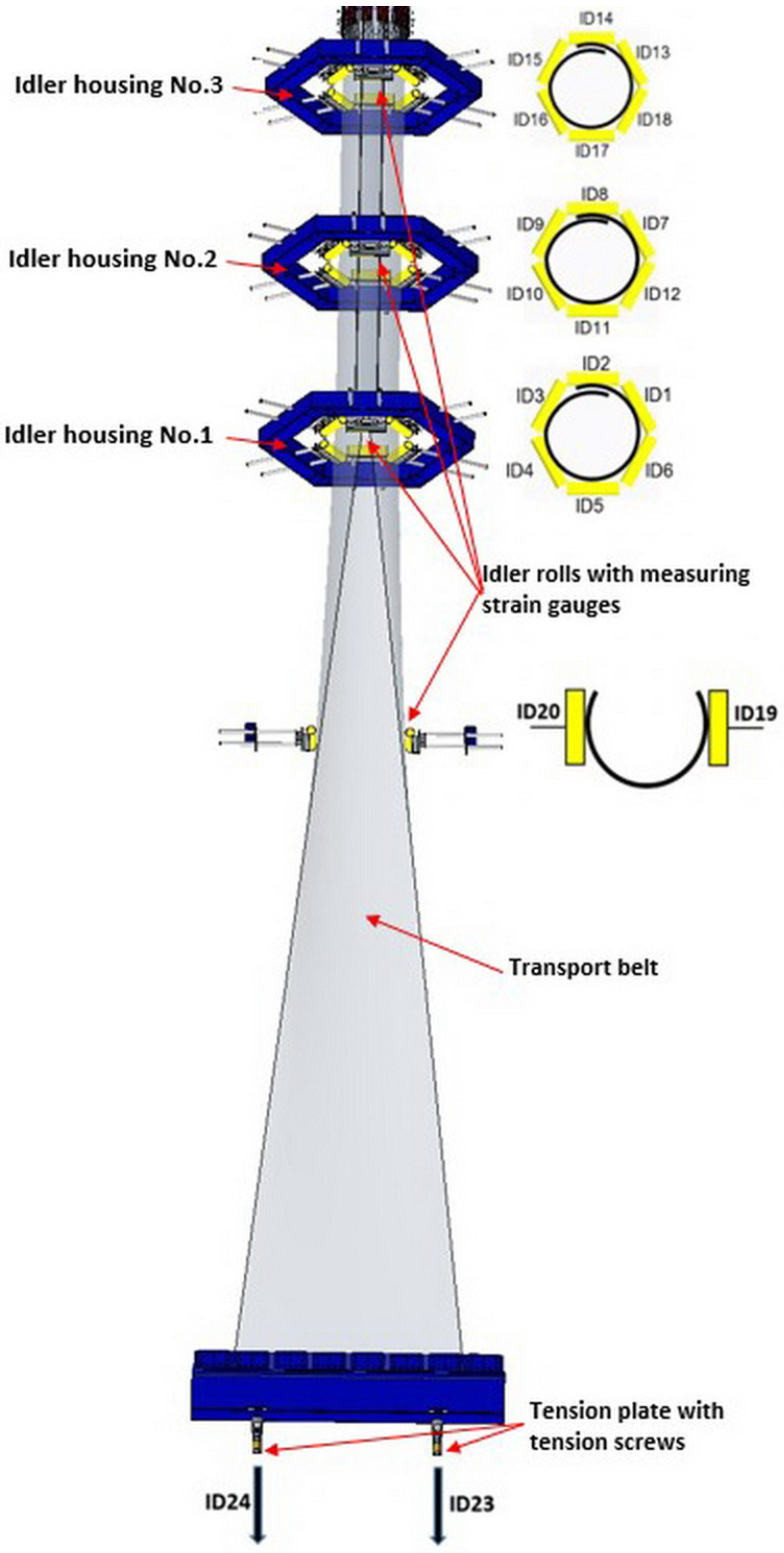
Principal scheme of the test rig at the Technical University of Kosice (Kosice, Slovakia).

**Figure 4 sensors-24-03810-f004:**
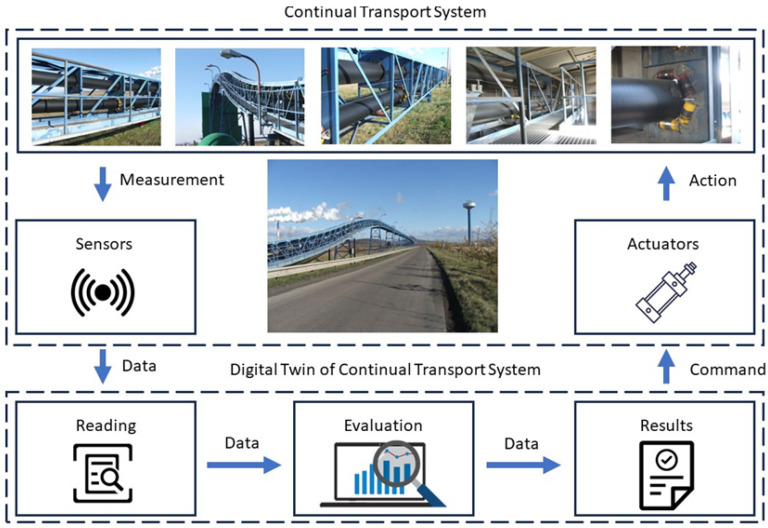
The principal structure of the “digital twin” of a continuous transport system.

**Figure 5 sensors-24-03810-f005:**
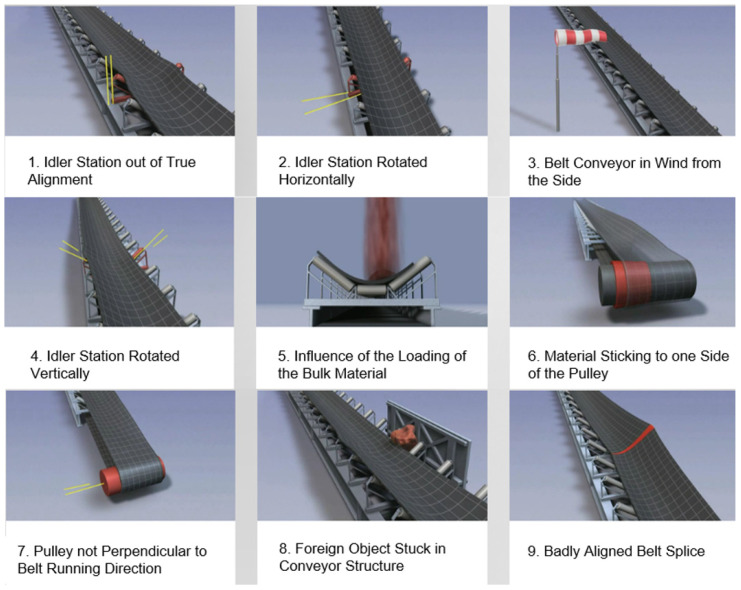
Causes of belt mistracking [21].

**Figure 6 sensors-24-03810-f006:**
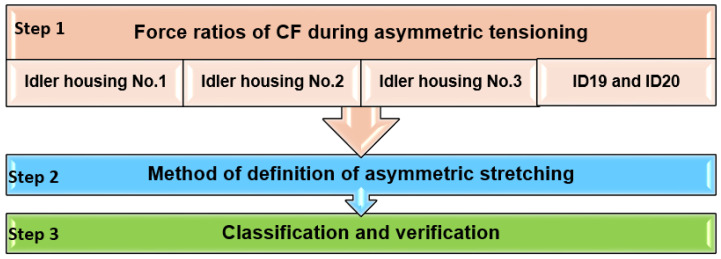
Processing of measured data when estimating the asymmetry of the transport belt’s tensioning in the test rig without material.

**Figure 7 sensors-24-03810-f007:**
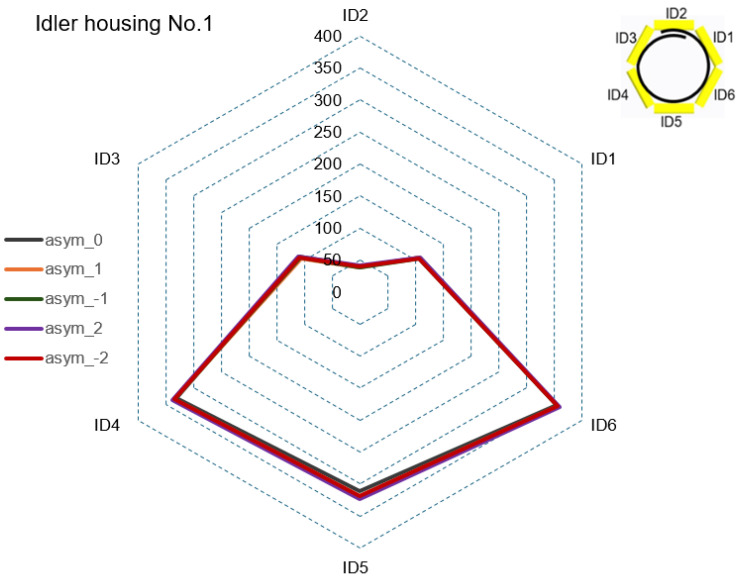
CF in idler housing No. 1 for all levels of set TF asymmetry.

**Figure 8 sensors-24-03810-f008:**
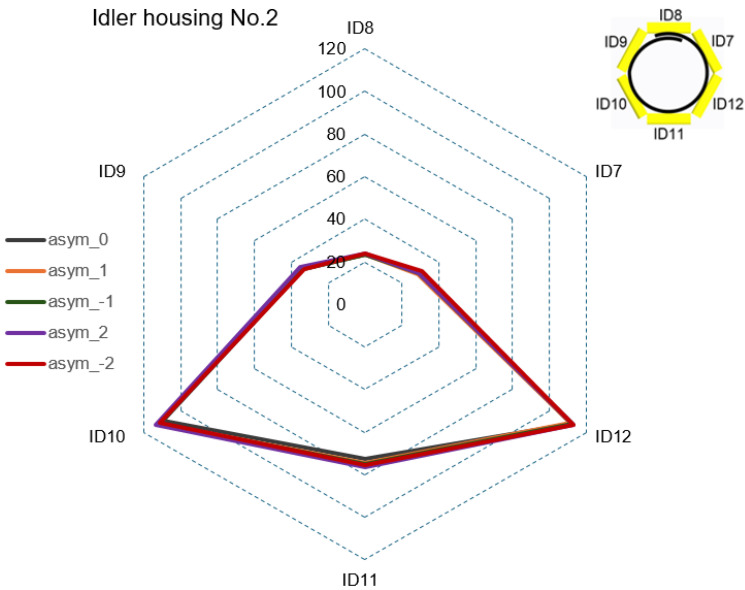
CF in idler housing No. 2 for all levels of set TF asymmetry.

**Figure 9 sensors-24-03810-f009:**
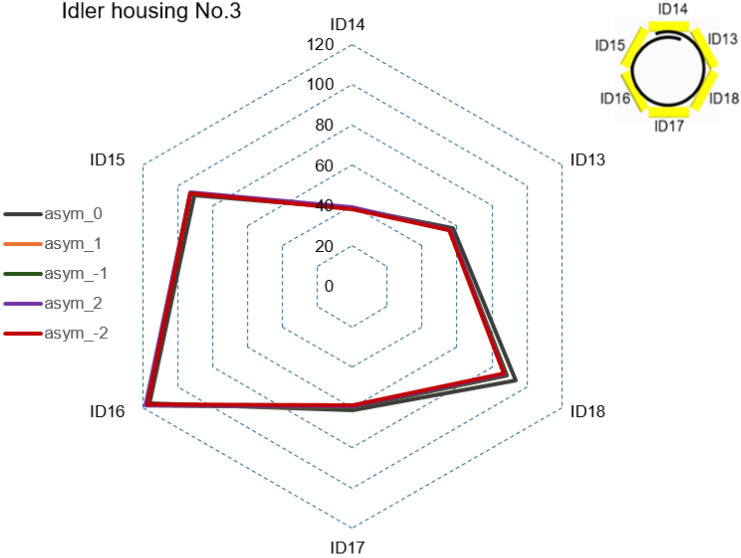
CF in idler housing No. 3 for all levels of set TF asymmetry.

**Figure 10 sensors-24-03810-f010:**
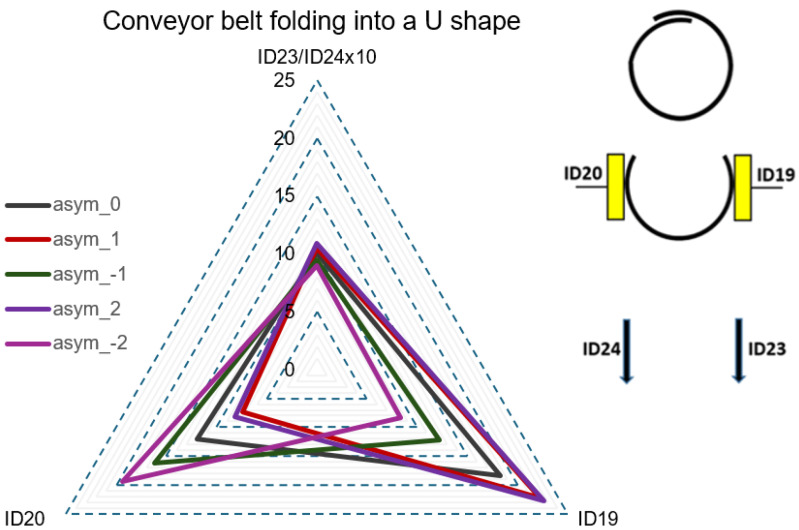
CF at the point of the conveyor belt folding into a U shape for all levels of the set TF asymmetry.

**Figure 11 sensors-24-03810-f011:**
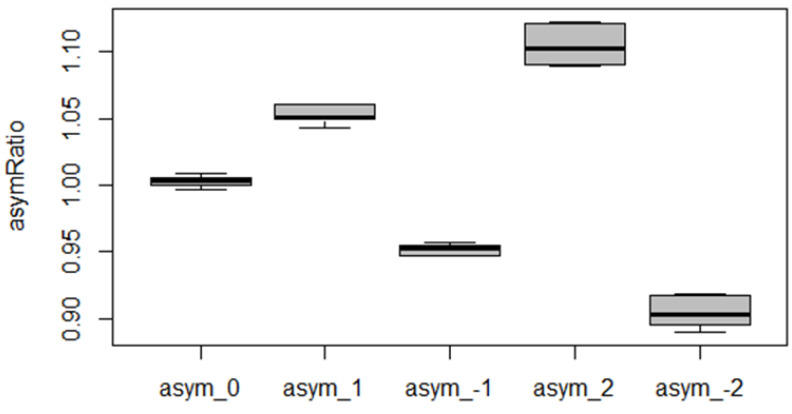
Box plots for the variable AsymRatio illustrate all the asymmetry levels set during the experiment.

**Figure 12 sensors-24-03810-f012:**
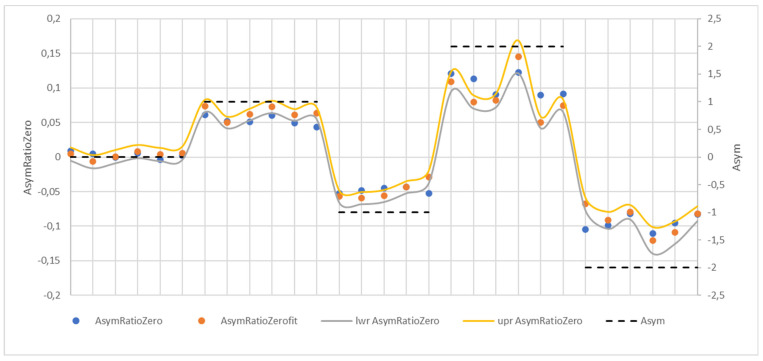
Verification of the regression model by comparing theoretical and empirical values.

**Figure 13 sensors-24-03810-f013:**
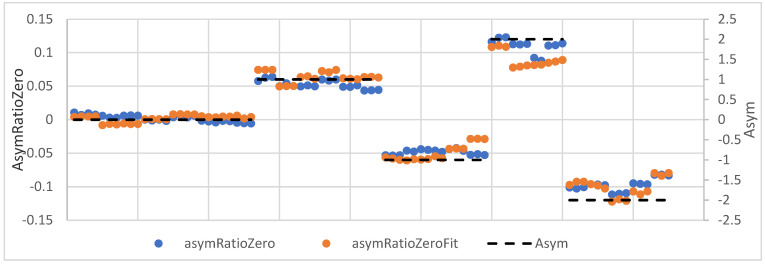
Verification of the regression model using independent test data.

**Figure 14 sensors-24-03810-f014:**
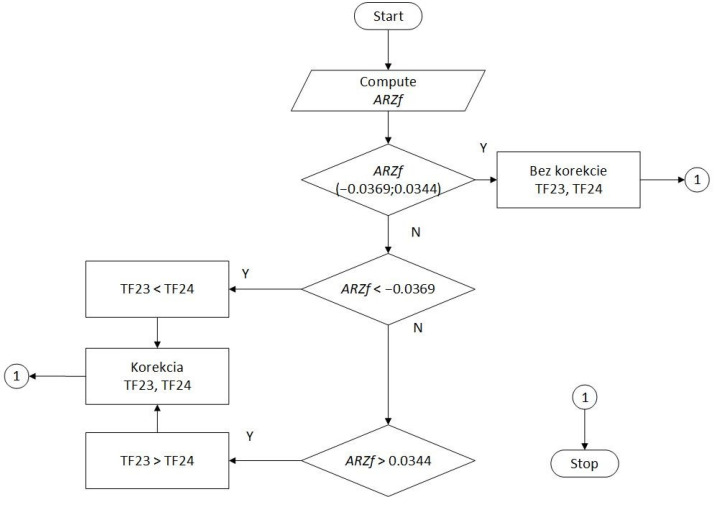
Evaluation classification algorithm for the belt mistracking measurement, using the *ARZf* variable.

**Table 1 sensors-24-03810-t001:** Selected settings for the asymmetry research in the conveyor belt’s tensioning without transported material in the test rig.

Set Tension Force [N]	Set Asymmetry Level (Difference TF23–TF24 [N] in ID23, ID24)
36,000	asym_0 (0 N)
40,000	asym_1 (1000 N)
44,000	asym_-1 (−1000 N)
	asym_2 (2000 N)
	asym_-2 (−2000 N)

**Table 2 sensors-24-03810-t002:** CF values [N] for all levels of set asymmetry at TF = 40,000 N.

Position	Asymmetry Type					Statistical Indicators				
	asym_0	asym_1	asym_-1	asym_2	asym_-2	Min	Max	Range	Average	Ratio
ID1	108	107	106	109	106	106	109	3	107	3%
ID2	39	39	39	41	39	39	41	2	40	5%
ID3	110	105	107	111	109	105	111	5	108	5%
ID4	332	334	334	338	334	332	338	6	334	2%
ID5	311	318	319	323	318	311	323	12	318	4%
ID6	354	356	356	359	356	354	359	5	356	1%
ID7	30	28	29	29	31	28	31	3	30	9%
ID8	23	23	24	24	24	23	24	1	24	4%
ID9	33	34	34	35	33	33	35	2	34	6%
ID10	110	112	112	113	111	110	113	4	112	3%
ID11	73	74	75	77	76	73	77	4	75	5%
ID12	112	111	112	113	113	111	113	2	112	2%
ID13	58	56	56	56	56	56	58	2	56	4%
ID14	39	39	39	39	38	38	39	1	39	2%
ID14	39	39	39	39	38	38	39	1	39	2%
ID15	90	93	93	93	93	90	93	3	92	3%
ID16	116	118	118	118	117	116	118	2	117	2%
ID17	61	60	60	60	59	59	61	2	60	4%
ID18	94	89	88	88	87	87	94	7	89	8%
ID19	18	22	12	23	8	8	23	14	17	85%
ID20	12	7	16	8	19	7	19	12	13	95%
ID23	19,846	20,218	19,383	20,713	18,813	18,813	20,713	1899	19,795	10%
ID24	19,757	19,270	20,358	19,006	20,879	19,006	20,879	1873	19,854	9%

**Table 3 sensors-24-03810-t003:** Tukey HSD criteria for pairwise comparison of all pairs.

Asymmetry	*p adj*
asym_1–asym_0	1 × 10^−7^
asym_-1–asym_0	1 × 10^−7^
asym_2–asym_0	0
asym_-2–asym_0	0
asym_-1–asym_1	0
asym_2–asym_1	0
asym_-2–asym_1	0
asym_2–asym_-1	0
asym_-2–asym_-1	4 × 10^−7^
asym_-2–asym_2	0

**Table 4 sensors-24-03810-t004:** Evaluation of the model parameters’ statistical significance.

Parameter	Estimate	Std. Error	t-Value	Pr(>|t|)
*a* _0_	–0.068193	0.009131	–7.468	6.27 × 10^−8^
*a* _1_	0.088633	0.021157	4.189	0.000285
*a* _2_	0.027980	0.005911	4.733	7.9 × 10^−5^

**Table 5 sensors-24-03810-t005:** Classification table.

	TRUE	
Classified	0	1
0	27	3
1	0	64

## Data Availability

The original contributions presented in the study are included in the article, further inquiries can be directed to the corresponding author.

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
