# Peer review of "Design of Evaluation Classification Algorithm for Identifying Conveyor Belt Mistracking in a Continuous Transport System’s Digital Twin"

_sensors, 2024, doi:10.3390/s24123810_

Round 1

Reviewer 1 Report

Comments and Suggestions for Authors

The paper presents an evaluation algorithm base on experimental measurement for identifing belt misalignment in continuous transportation systems. There are still some minor issues that need to be addressed in the paper.

The titles of section 3 and 4 do not reflect the content. It is recommended to modify the titles of parts 3 and 4.

In section 5,Suggest adding the limitations of the evaluation algorithm and the issues that need to be addressed in practical applications.

Reviewer 2 Report

Comments and Suggestions for Authors

1)In Part 1 Introduction, a detailed point-by-point account of the contributions made in this paper would have made the contributions clearer.

2)Is Digital Twin missing a subheading on line 121 of this article.

3)The character in Figure 5 should be clearer.

4)Formula numbers should align.

5)An outlook can be added at the end of the paper to further discuss possible future work and solutions.

6)Reference formats need to be consistent.

Comments on the Quality of English Language

The English needs further improvement.

Reviewer 3 Report

Comments and Suggestions for Authors

Article:

Design of Evaluation Classification Algorithm for Identifying Belt Mistracking in the Continuous Transport System’s Digital Twin

Expertise:

Belt Mistracking is an undesirable condition that can cause the conveyor belt to converge and thus seriously disable the entire transport system.

Within this paper, the research is presented, aiming to verify the hypothesis that based on a measurement of selected parameters, it is possible to identify Belt Mistracking in a continuous transport system. The research results confirmed the established hypothesis. An evaluation algorithm was created to be used for on-time evaluation. The proposed algorithm is also suitable for the needs of a digital twin of a continuous transport system.

Overall, the text is very well-written and comprehensive. This paper is recommended for publication, minor suggestions for improvement are given below.

Comments and suggestions:

1. In paper title, I suggest that it says “… Identifying Conveyor Belt Mistracking …” to add clarity.

2. Page 3, Figure 1: For Figure 1 two icons have the same caption: “Measurement System” – I suggest that another descriptive word is added so that it is possible to discern the differences between these two systems.

3. Page 7, line 200: “The experiments were to find” – a verb is seemingly missing in passive voice.

4. Page 9, Figure 7-10: I suggest that an explanation is given for the scales of the strain gauges (IDs), as they are different in every figure.

5. Page 12, line 281 – ANOVA should be spelled out and the test should be explained.

6. Page 12, Figure 11: I suggest that the explanation of the Figure is extended in the Figure’s caption.

Comments on the Quality of English Language

Given above.

Round 2

Reviewer 2 Report

Comments and Suggestions for Authors

I have carefully reviewed the changes and responses you have made in response to my previous review comments and believe that they effectively address the issues previously noted. The paper has been significantly improved in terms of structure, content, logic and formatting.

I appreciate the effort and seriousness with which you have done this.

Comments on the Quality of English Language

I think the author's English is okay, but there is still room for improvement.

Author Response

We accept the reviewer’s comment. The manuscript was checked by the english lector.
